# From Complex to Simple: Unraveling the Cognitive Tree for Reasoning with Small Language Models

**Junbing Yan[1], Chengyu Wang[2], Taolin Zhang[2], Xiaofeng He[1],**
**Jun Huang[2], Wei Zhang[13*]**

[1] School of Computer Science and Technology, East China Normal University
[2] Alibaba Group
[3] Shanghai Institute for AI Education

{junbingyan531,zhangwei.thu2011}@gmail.com, hexf@cs.ecnu.edu.cn
{chengyu.wcy,zhangtaolin.ztl,huangjun.hj}@alibaba-inc.com

## Abstract

Reasoning is a distinctive human capacity, enabling us to address complex problems by breaking them down into a series of manageable cognitive steps. Yet, complex logical reasoning is still cumbersome for language models. Based on the dual process theory in cognitive science, we are the first to unravel the cognitive reasoning abilities of language models. Our framework employs an iterative methodology to construct a **Cognitive Tree** (CogTree). The root node of this tree represents the initial query, while the leaf nodes consist of straightforward questions that can be answered directly. This construction involves two main components: the implicit extraction module (referred to as the intuitive system) and the explicit reasoning module (referred to as the reflective system). The intuitive system rapidly generates multiple responses by utilizing in-context examples, while the reflective system scores these responses using comparative learning. The scores guide the intuitive system in its subsequent generation step. Our experimental results on two popular and challenging reasoning tasks indicate that it is possible to achieve a performance level comparable to that of GPT-3.5 (with 175B parameters), using a significantly smaller language model that contains fewer parameters (<=7B) than **5%** of GPT-3.5. [1]

## 1 Introduction

The human brain is akin to a garden, where instincts are seeds that sprout and grow, while reason acts as the gardener, pruning and nurturing the plants of knowledge to bloom into the flowers of enlightenment. For machines, recently, Large Language Models (LLMs) have demonstrated their abilities to tackle diverse tasks through instantaneous question answering, exhibiting some levels of intelligence (Ouyang et al., 2022; Wei et al., 2022a; Wang et al., 2022b).

However, to cross the chasm between machines and humans, three main challenges still lie ahead: **1)** Reasoning ability. When it comes to mathematical and reasoning problems, the performance of LMs is still not satisfactory (Koyejo et al., 2022; Cobbe et al., 2021b). **2)** Cognition capacity. The evaluation and decision-making process regarding the problem and its current state is of paramount importance, especially when dealing with problems that involve lengthy reasoning chains or multi-step solutions. However, current methods (Wei et al., 2022b; Yao et al., 2023) often lack comprehensive validation and tend to focus on verifying intermediate results (Imani et al., 2023). **3)** Efficiency. The deployment and inference costs of LLMs are relatively high, especially when utilizing parameter-free inference enhancement techniques (Wei et al., 2022b; Yao et al., 2023). These techniques require extensive contexts and multiple steps of answer generation, leading to a further increase in inference costs and time.

We suggest that valuable insights into addressing these challenges can be derived from the cognitive processes of humans. In cognitive science, the *dual process theory* (Evans, 1984, 2003, 2008; Sloman, 1996) states that our brain initially employs an implicit, unconscious, and intuitive process known as the **Intuitive System**, which retrieves relevant information. This is followed by an explicit, conscious, and controllable reasoning process called the **Reflective System**. The Intuitive System is capable of providing resources in response to requests, while the Reflective System facilitates a deeper exploration of relational information through sequential thinking in the working memory. Although slower, the Reflective System possesses a unique human rationality (Baddeley, 2010). In complex reasoning tasks (including logical reasoning and mathematical reasoning tasks),

---

*\* Correspondence to Wei Zhang.*

[1]The source code will be released in the EasyNLP framework (Wang et al., 2022a). URL: https://github.com/alibaba/EasyNLP

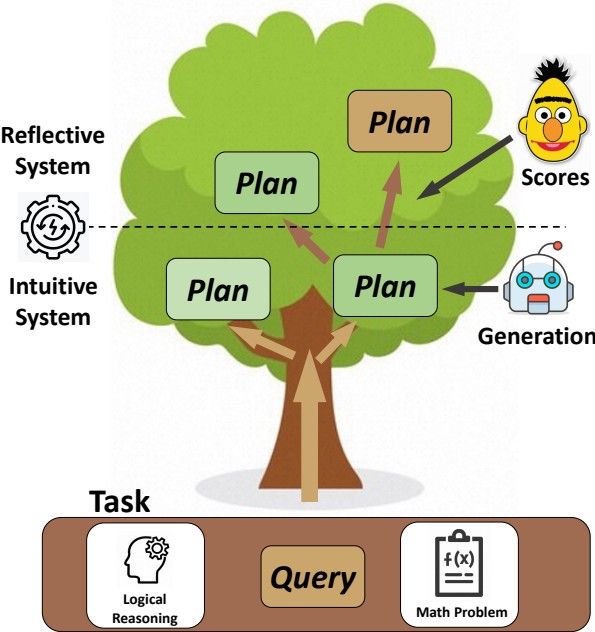

Figure 1: A schematic illustration of the proposed framework named CogTree. An intuitive system is employed to generate candidate plans, while a reflective system verifies the plausibility of each plan to guide the next generation of the intuitive system. This iterative process is repeated to create the tree structure for reasoning.

these two systems coordinate with each other, engaging in iterative cycles of *fast and slow* thinking (Daniel, 2017).

In this paper, we propose the **Cognitive Tree** (CogTree) framework to address the aforementioned issues. Inspired by the dual process theory, our system consists of the Intuitive System and the Reflective System. In our implementation, the Intuitive System and the Reflective System are both generative models, albeit with distinct objectives. The Intuitive System employs in-context examples to dissect intricate problems into sub-problems and produce responses to the query. Conversely, the Reflective System evaluates the outcomes generated by the Intuitive System and chooses the most likely solution to provide guidance for the next generation step. The aforementioned process is an iterative tree generation process, which continues until the original problem is decomposed into manageable sub-problems (corresponding to nodes on the tree) that can be easily solved.

Our main contributions are as follows:

- **Problem Decomposition Paradigm.** We propose a novel framework based on human cognition called Cognitive Tree (CogTree) for

solving complex reasoning problems.

- **Improved Validation Capabilities.** By exposing the model to contrastive examples of correct decisions versus incorrect or ambiguous ones, we can improve the model's ability to make decisions (cognition ability). Additionally, apart from evaluating the model's judgment of intermediate results, we have also integrated the model's assessment of the overall correctness of the reasoning process.

- **Efficient Framework.** The combination of the Intuitive System and the Reflective System can be applied to various reasoning tasks, i.e., both logical reasoning and mathematical reasoning. Notably, we are able to attain comparable reasoning performance to models with substantially larger parameter sizes, such as GPT-3.5 with 175B parameters, while utilizing relatively small language models (with 1.5B and 7B parameters). This allows the trained small models to be deployed online for efficient inference.

## 2 Cognitive Tree (CogTree) Framework

The reasoning ability of humans primarily arises from acquiring pertinent information from the environment and subconscious processing (Prystawski and Goodman, 2023). In our approach, we incorporate a tree structure to systematically tackle reasoning problems, taking inspiration from human problem-solving procedures. This methodology aligns with the planning processes analyzed by Newell et al., 1959; Newell and Simon, 1972. Newell and his colleagues defined problem solving (Newell et al., 1959) as the exploration of a combinatorial problem space, represented as a tree.

In our mathematical and logical reasoning setting, each node $n$ in the *cognitive tree* $\mathcal{T}$ represents either a theory in a logical set or the solution to a sub-problem in a mathematical question. An edge $e$ of the tree corresponds to the evaluation of the current node's state $s$, which can be a confidence score or a classification result. The problem decomposition module (i.e., the Intuitive System) receives the theory and the original problem as input, and generates the decomposition of the original problem. Next, the newly generated nodes are used to expand the tree, providing the reasoning module (i.e., the Reflective System) with the information that needs to be verified.

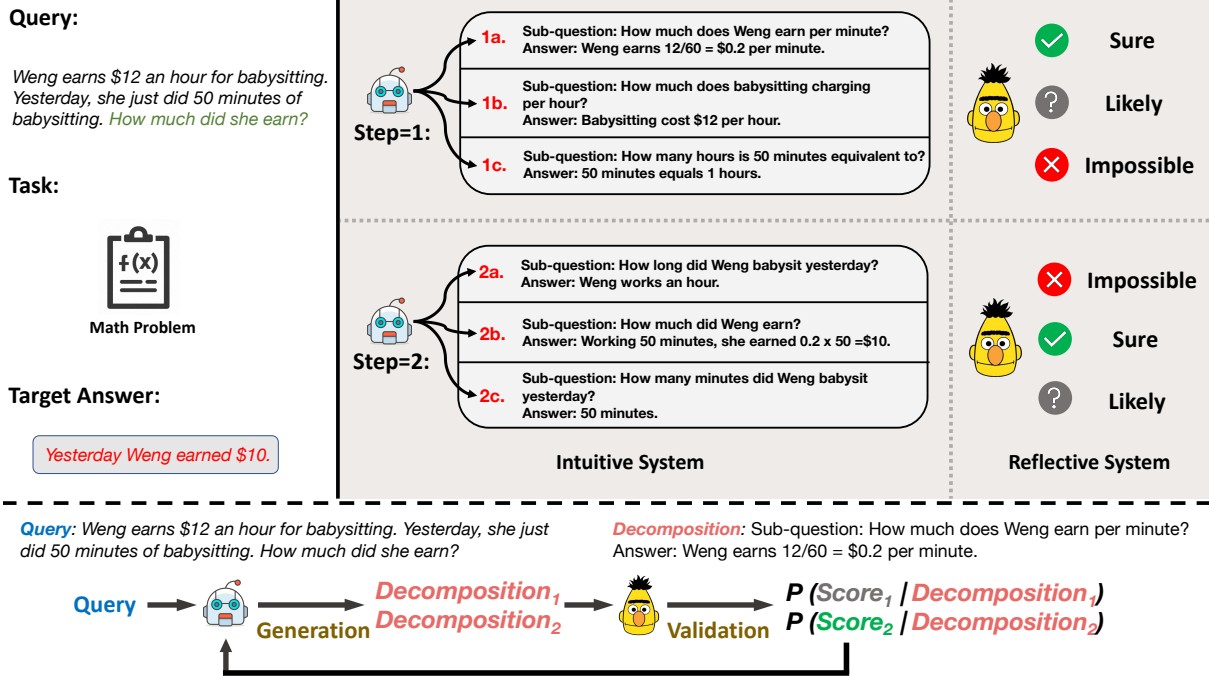

Figure 2: An illustration of how the Intuitive System and the Reflective System work to incrementally produce a mathematical reasoning problem. At each step, the Intuitive System generates a set of decompositions based on the query. The Reflective System then scores the candidate decompositions and returns the top-ranked ones. The process terminates when the decomposition successfully matches the target answer.

The discernment capacity of the Reflective System plays a pivotal role in enhancing the overall efficacy of the model (Imani et al., 2023). In particular, we utilize the cross-checking technique to not only verify the precision of intermediate outcomes but also validate the accuracy of the entire reasoning process upon its completion. However, relying solely on individual decision judgments for training the model is inadequate (Imani et al., 2023). To augment the model's ability to evaluate the state $s$, we propose the implementation of a comparative reinforcement approach. This approach entails introducing a new training objective, whereby the model is tasked with maximizing the disparity in vector space between representations of correct decisions and representations of incorrect or ambiguous decisions.

In addition, efficiency pertains to the efficacy of the problem decomposition and verification process. Generating lengthy texts using a model as massive as 175B entails substantial time and financial expenses. To tackle this issue, our method can be implemented by simply fine-tuning a comparatively smaller model (1.5B or 7B) exclusively for reasoning tasks. This enables us to deploy the model for these tasks with minimal costs.

## 3 Implementation

In this section, we describe the implementations of CogTree in detail.

### 3.1 Intuitive System

The generative capability of the Intuitive System serves as the foundation for constructing the Cognitive Tree. Thus, we choose decoder-only models (e.g., GPT2-XL (Radford et al., 2019) or LLaMA-7B (Touvron et al., 2023)) as the Intuitive System.

To enhance the effectiveness of the Intuitive System, we employ an in-context approach. Let us define the Query ($\mathcal{Q}$) as the ultimate goal for logical reasoning problems or the question to be answered in mathematical problems. In the case of logical reasoning problems, the Decomposition ($\mathcal{D}$) involves further breaking down the goal into smaller components, where reasoning through this decomposition enables the attainment of the goal. For mathematical problems, it refers to one of the sub-problems derived from the original problem, and solving this sub-problem contributes to resolving the original problem as a whole. The Decomposition set ($Z$) represents the collection of decompositions for all examples in the training set. In our approach, we retrieve $K$ examples (e.g., Query: $\mathcal{Q}$;

Step=1:
**Query₁**: earth rotating on its axis causes cycles of day and night on earth
**Decomposition₁**: [ *earth is a planet that rotates on its tilted axis* ] + [ *a planet rotating causes cycles of day and night on that planet* ]

Step=2:
**Query₂**: earth is a planet that rotates on its tilted axis
**Decomposition₂**: [ *the earth rotates on its tilted axis* ] + [ *earth is a kind of planet* ]

Figure 3: Example of Query and Decomposition for logical reasoning.

Step=1:
**Query₁**: Natalia sold clips to 48 of her friends in April, and then she sold half as many clips in May. How many clips did Natalia sell altogether in April and May?
**Decomposition₁**: *How many clips did Natalia sell in May? Natalia sold 48/2 = 24 clips in May.*

Step=2:
**Query₂**: Natalia sold clips to 48 of her friends in April, and then she sold half as many clips in May. How many clips did Natalia sell altogether in April and May? Decomposition₁: How many clips did Natalia sell in May? Natalia sold 48/2 = 24 clips in May.
**Decomposition₂**: *How many clips did Natalia sell altogether in April and May? Natalia sold 48+24 = 72 clips altogether in April and May.*

Figure 4: Example of Query and Decomposition for mathematical problems.

Decomposition: $\mathcal{D}$) from the inference decomposition set ($Z$), which are then utilized as the context for the model's input. Detailed examples are shown in Figure 3 and Figure 4.

Then, the output can be generated as $y \sim f_\theta(y|x, z_{1\cdots K})$. Here, $z$ represents the $K$ examples[2] recalled from the decompositions set $Z$, where $Z = \{z_1, \cdots, z_L\}$. In practice, we use the Intuitive System to obtain the representation of the current query (final transformer block's activation) and calculate the cosine similarity with the representations of other queries in set $Z$. We then retrieve the $K$ most similar queries from the set. Moreover, $[y] \sim f_\theta(y|x, z_{1\cdots K})$ is sampled as a continuous language sequence.

## 3.2 Reflective System

The Reflective System differs from the Intuitive System in terms of its approach of generating insights. While the Intuitive System relies on quick intuition, the Reflective System's role is to evaluate the decompositions to determine their acceptability. In practice, we employ two methods to verify the results: the verification of intermediate processes and the verification of the entire reasoning chain.

---

[2]In the experiment, we set $K = 5$.

Given the current state $s$ (Query: $\mathcal{Q}$ with Decomposition: $\mathcal{D}$), we utilize the Reflective System, which shares the same model architecture as the Intuitive System, to generate a score $v$ that validates the current state. This is represented by $V(f_\theta, s) \sim f_\theta(v|s)$. Additionally, based on the complete reasoning chain $S = \{s_1, \cdots, s_i, \cdots, s_n\}$, we employ the Reflective System to produce an overall score $o$, which can be expressed as $O(f_\theta, S) \sim f_\theta(o|S)$.

To generate the scores $v$ and $o$, we utilize the model to produce a classification result. Since the answers generated by the Intuitive System can sometimes be misleading and cannot be accurately assessed at this stage, we adopt a prompt-based approach and treat it as a classification problem, where the model outputs one of three categories: *sure*, *impossible*, or *likely*. The *likely* response signifies that the generated answer is plausible but requires further verification.

## 3.3 Training

**Intuitive System.** Supervised Fine-tuning (SFT) has demonstrated its effectiveness in aligning with human intentions (Ouyang et al., 2022). In our approach, the Intuitive System is designed to decompose the queries (i.e., complex problems) into sub-problems by leveraging in-context examples. Since we employ generative models as our Intuitive System, the loss calculation is only necessary for the generative text (without the given context) during auto-regressive computation. Given a sample of tokens with a length of $N$ denoted as $X$, where $X = \{x_1, \cdots, x_i, \cdots, x_n\}$. Furthermore, we define the sequence length of in-context examples as $M$. We use a standard language modeling objective to maximize the following likelihood function:

$$\mathcal{L}_{\mathcal{IS}} = \sum_{i>M}^{N} log\, P(x_i|x_1, \cdots, x_{i-1}; \theta) \quad (1)$$

**Reflective System.** The acquisition of the Reflective System can be achieved through the same training approach as the Intuitive System, which involves utilizing positive and negative samples to obtain classification results from the model. Since the Reflective System is primarily focused on generating judgments for a given state $s$, the loss function can be defined as follows:

$$\mathcal{L}_{\mathcal{RS}} = log\, P(v|s; \theta) \quad (2)$$

| Dataset | Input | Output |
|---------|-------|--------|
| **EB** | A hypothesis that needs to be proven and a set of theory. (Hypo: Phobos is a kind of moon. Theory: [Mars is a kind of planet; moons orbit planets; Phobos orbits Mars.]) | Yes or No to indicate whether or not the hypothesis can be proven based on the theory. (Yes) |
| **GSM8K** | A math word problem (Natalia sold clips to 48 of her friends in April, and then she sold half as many clips in May. How many clips did Natalia sell altogether in April and May?) | A number denoting the solution to the mathematical problem. (72) |

Table 1: Task overview. Examples of input and output are printed in blue.

| Dataset | Split | # Samples | Max Steps | Avg Steps |
|---------|-------|-----------|-----------|-----------|
| EB | Train | 1313 | 17 | 3.2 |
| | Dev | 187 | 15 | 3.2 |
| | Test | 340 | 11 | 3.3 |
| GSM8K | Train | 6726 | 9 | 3.6 |
| | Dev | 747 | 8 | 3.5 |
| | Test | 1319 | 11 | 3.7 |

Table 2: Data statistics of EB and GSM8K.

However, the effectiveness of this training method is found to be unsatisfactory. In cognitive theory, human decision-making behavior arises from the comparative analysis of various options (Festinger, 1957). Drawing inspiration from the cognitive theory, we adopt a contrastive learning approach to enhance the model's ability to distinguish between different states. The fundamental concept of contrastive learning is to learn representations of positive and negative samples by maximizing their distance in the sample space (Chen et al., 2020). Consequently, the selection of negative samples plays a critical role in determining the effectiveness of contrastive learning.

For logic reasoning datasets, one approach to generate more challenging negative examples is to replace one of the theories in decomposition with another theory from the current theory set. This negative example is more challenging for the model to distinguish because the theories within the same theory set are more similar.

For mathematical problems, since our experimental dataset GSM8K (Cobbe et al., 2021a) only provides the correct answer itself, it does not offer incorrect solutions. We use the dataset PRM800K (Lightman et al., 2023) to enhance the learning process, where there are ambiguous responses $S' = \{s'_1, \cdots, s'_i, \cdots, s'_n\}$ (seemingly correct but actually incorrect). The judgment generated by Reflective System is $V' =$

$\{v'_1, \cdots, v'_i, \cdots, v'_n\}$. By maximizing the distance between $v'$ and the correct answers $v$, we can enhance the learning process. Let $g(v', \cdot)$[3] be a matching function between negative sample $v'$ and the positive sample $v$. The loss function for contrastive learning can be expressed as follows:

$$\mathcal{L}_{\mathcal{CL}} = \frac{\exp(f(v, y))}{\exp(f(v, y)) + \sum_{v' \sim V'} \exp(f(v, v'))} \quad (3)$$

Hence, the total loss function for the Reflective System is given by:

$$\mathcal{L}_{\text{total}} = \lambda \cdot \mathcal{L}_{\mathcal{RS}} + (1 - \lambda) \cdot \mathcal{L}_{\mathcal{CL}} \quad (4)$$

Here, $\lambda$ is the hyper-parameter. We conduct experiments on $\lambda$ and choose $\lambda = 0.5$ as the best setting.[4]

## 4 Experiments

We perform an extensive evaluation of our method utilizing two well-established benchmark datasets: the Entailment Bank (EB) (Dalvi et al., 2021) and GSM8K (Cobbe et al., 2021a). EB consists of human-annotated tuples containing information about theories, provable goals, and corresponding reasoning paths. GSM8K, on the other hand, presents a challenging arithmetic reasoning task that language models frequently find difficult to tackle (Hendrycks et al., 2021; Cobbe et al., 2021b). Examples of these two datasets are in Table 1. The dataset statistics are shown in Table 2.

### 4.1 Experimental Setup

In our experiments, we use GPT2-XL (Radford et al., 2019) and LLaMA-7B (**?**) from the Huggingface transformers (Wolf et al., 2020) library as the underlying models for the Intuitive System and the Reflective System.

---

[3]In the experiment, we use cosine similarity as $g(v', \cdot)$
[4]For specific details, please refer to the Section 4.7.

| Model | #Params. | EB Accuracy (%) | Δ (%) | GSM8K Accuracy (%) | Δ (%) |
|---|---|---|---|---|---|
| **Comparative Systems** | | | | | |
| GPT-3.5 (code-davinci-002) | 175B | 80.76 | - | 16.17 | - |
|   + Standard prompt | 175B | 84.23 | +3.47 | 17.03 | +0.86 |
|   + Chain-of-thought prompt | 175B | 92.45 | +11.69 | 60.27 | +44.10 |
|   + Tree-of-thought prompt | 175B | 93.31 | +12.55 | **61.39** | +45.22 |
| **Our Models (CogTree)** | | | | | |
| GPT2-XL (Intuitive System only) | 1.5B | 82.37 | - | 23.53 | - |
|   + GPT2-XL (as Reflective System) | 1.5B | 92.63 | +10.26 | 35.84 | +12.31 |
|   + LLaMA (as Reflective System) | 7B | 93.16 | +10.79 | 34.68 | +11.15 |
| LLaMA (Intuitive System only) | 7B | 86.14 | - | 43.52 | - |
|   + GPT2-XL (as Reflective System) | 1.5B | 93.25 | + 7.11 | 47.80 | +4.28 |
|   + LLaMA (as Reflective System) | 7B | **94.25** | +8.11 | 61.28 | +17.76 |

Table 3: Overall test set performance in terms of accuracy and relative improvement.

During training, we use the Adam optimizer with $\beta_1 = 0.9, \beta_2 = 0.999, \epsilon = 1e-8$. We use the learning rate $\gamma = 1e-4$ for both Systems. We use a batch size of 4 and set a large epoch number (i.e., 100) and use the validation set to do early stopping. In practice, the best epoch is often within 50. During the inference stage, in each step, we use the Intuitive System to generate the top_beam=3 answers and then let the Reflective System select the most probable answer to continue generating for the next step. We add an "end" marker. The inference process stops when the Intuitive System generates the "end" marker or when the maximum number of inferences, which is 20, is reached. For each Query, we perform 5 complete reasoning process.[5]

Following Zhou et al., 2022, we use code-davinci-002[6] (code-davinci-002 is constructed on the foundation of the GPT-3.5 architecture) for comparative experiment due to its strong reasoning abilitiy and employ various prompt strategies to conduct our experiments on GPT2-XL and LLaMA-7B. For each example, we sample Standard prompting (detailed cases to be described in Section 4.2) and Chain-of-thought (CoT) prompting for 100 times for average performance. For Tree-of-thought (ToT), at each step we generate 5 candidate answers and sample values 3 times for each example.

### 4.2 Results on Entailment Bank

On EB, followed by Zhao et al., 2023, we assess the capabilities of systems in distinguishing between provable and non-provable goals. To accomplish this, we assign a non-provable goal to each development and testing theories by selecting it from other (theory, goal, reasoning path) samples. The selection is adversarial: We input all the goals in the set into our pre-trained model separately, obtaining the last output of the last transformer block as the representation of each goal. We proceed by computing the cosine similarity between all non-provable goals and the provable goal. Based on this computation, we identify the hard negative example with the highest similarity. For a given theory $T$ and a query $Q$, we allow the system to generate a reasoning path $S$ and obtain the proof value $o = f_\theta(o|S)$ for that path. Given the choices "Sure/Likely/Impossible", we say "$Q$ is provable" when the value $o$ is "sure" and not provable otherwise.

**Baselines. Vanilla**: The raw input to the model is as follows: Query: $\mathcal{Q}$; Theory Set: $\mathcal{T}$, followed by a question: Based on the theory, can the goal be approved? Please answer with yes or no. **Standard prompt**: We use an input-output (IO) prompt with 5 in-context examples (e.g. Query: $\mathcal{Q}$; Theory Set: $\mathcal{T}$; Answer: $\mathcal{A}$.). **Chain-of-though prompt**: For chain-of- thought (CoT) prompting, we augment each input-output pair using examples with complete reasoning chain (e.g. Query: $\mathcal{Q}$; Theory Set: $\mathcal{T}$; Reasoning Chain: $\mathcal{S}$; Answer: $\mathcal{A}$.). **Tree-of-thought prompt**: In each step, we provide the model with the theory set and ask the model to select two theories from the set to generate a new inference, which is added to the theory set while removing the two selected theories. Each step generates five candidates, and the model gen-

---

[5]Detailed examples are shown in Figure 7 and Figure 8.
[6]https://openai.com/

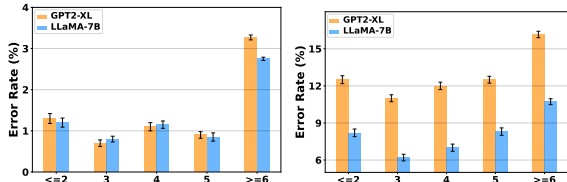

Figure 5: Samples failed at each step on EB (left) and GSM8K (right).

erates subsequent inferences based on the highest likelihood until the set ultimately contains only one inference, which is compared with the goal to determine consistency (e.g., theory$_i$ + theory$_j$ -> inference (theory set without theory$_{i\&j}$)).

**Main results.** Table 3 presents the classification accuracy on EB. The results demonstrate that GPT2-XL (1.5B), trained exclusively on in-context examples, outperforms GPT-3.5 (175B) despite having fewer than 1% of the model's parameters. By incorporating CoT and ToT augmentation methods, the accuracy of GPT-3.5 is substantially enhanced, reaching 92-93%. Furthermore, when our approach is combined with the Reflective System for result verification, even higher performance is achieved (94% by LLaMA-7B), surpassing the prompt augmentation method employed by GPT-3.5.

**Error Analysis.** Although our model has achieved satisfactory results, it still exhibits deficiencies in certain cases. We have conducted an examination of instances where the model's inference has failed (Detailed examples are shown in Figure 5). Specifically, we observed a decline in the model's accuracy when the length of the inference chain exceeds 10 steps. Our model struggles to determine the appropriate decomposition of the goal in such cases. This limitation may arise from the excessively long and divergent nature of the chain required to reach the goal, which surpasses the model's current capacity for abstraction in this regard. It is worth noting that even for humans, this task can be complex, often requiring multiple attempts to arrive at the final inference chain.

### 4.3 Results on GSM8K

On GSM8K, we employ in-context examples to enable the model to generate sub-questions for a given problem. In the implementation, we conduct five sampling iterations at each decomposition step and choose the one with the highest probability for generating the next step. Once all the sub-questions have been generated, we proceed to have the model answer each sub-question sequentially, ultimately deriving the final answer.

**Baselines. Vanilla**: Directly input the question and let the model to answer (e.g. Query: $\mathcal{Q}$). **Standard prompt**: We use an input-output (IO) prompt with 5 in-context examples (e.g. Query: $\mathcal{Q}$; Answer: $\mathcal{A}$). **Chain-of-though prompt**: For CoT, we use the step-by-step solution of mathematical problems as enhanced input-output pair (e.g. Query: $\mathcal{Q}$; Reasoning Chain: $\mathcal{S}$; Answer: $\mathcal{A}$). **Tree-of-though prompt**: We adopt the methodology described in ToT (Yao et al., 2023). Our implementation involves the incremental generation of the answer, with the model producing one step at a time. We sample the results five times for each step and employ the model to assess the generated outcomes until the final answer is obtained[7].

**Main Results.** Experimental results are shown in Table 3. It is evident that the direct question-answering accuracy of GPT-3.5 is merely 16%. The traditional input-output approach did not improve its effectiveness. In contrast, CoT and ToT demonstrate a significant enhancement in solving mathematical problems, with an improvement of approximately 44%. This indicates the crucial role of providing the model with example reasoning chains for tasks involving multi-step inference. Our SFT-improved GPT2 achieves only 23.5% accuracy, which may be attributed to the scale low (Kaplan et al., 2020) caused by model parameters. The performance of our LLaMA-7B and GPT-3.5 models is nearly indistinguishable, suggesting that fine-tuning small models using our provided method can approach the performance of larger models. Notably, the inclusion of the Reflective System on GSM8K leads to a greater overall improvement compared to EB, indicating that using the Reflective System can yield better results, particularly in more complex problems.

**Error Analysis.** We have found that the examples where the model fails to answer can be divided into two categories. The first category is the failure in decomposing the question into subproblems (Detailed examples are shown in Figure 5), which is consistent with the findings of Zhou et al. (2022). Such failures can be resolved through manual decomposition. The second category of failure is

---

[7]It is worth noting that our tree construction method differs significantly from ToT (Yao et al., 2023). ToT employs a bottom-up approach, whereas we utilize explicit questioning to break down the problem for the model and address it in a top-down fashion, step by step.

| Model | EB | GSM8K |
|-------|-----|-------|
| **GPT-XL** | | |
| +Specialize | 81.54% | 21.43% |
| +DecomP | 83.27% | 24.35% |
| +Self-Ask | 82.69% | 25.46% |
| +CogTree | **93.16%** | **34.68%** |
| **LLaMA-7B** | | |
| +Specialize | 84.32% | 34.21% |
| +DecomP | 82.43% | 39.75% |
| +Self-Ask | 83.98% | 38.35% |
| +CogTree | **94.25%** | **61.28%** |

Table 4: Performance metrics for different methods with EB and GSM8K.

| Model | EB | GSM8K |
|-------|-----|-------|
| GPT-XL | **92.63%** | **35.84%** |
| - decomposition | 62.38% | 18.94% |
| LLaMA-7B | **93.25%** | **61.28%** |
| - decomposition | 69.79% | 27.16% |

Table 5: Ablation study in terms of F1.

when the model provides inaccurate answers to the subproblems. These failures are frequently observed in GPT2-XL and are the main reason for the unsatisfactory performance of GPT2. One possible reason for these failures is the inadequacy of the model's parameter size, which hinders its ability to acquire fundamental mathematical capabilities.

## 4.4 Backbone Modifications

We conduct experiments employing different backbones for the Intuitive System and the Reflective System. The results presented in Table 3 demonstrate that when GPT2-XL is utilized as the Intuitive System and LLaMA-7B as the Reflective System, there is an observed improvement in overall performance on the EB dataset, compared to using GPT2-XL as the Reflective System. However, when a more powerful model is employed as the Reflective System on the GSM8K dataset, there is no noticeable performance enhancement. This finding suggests that the Intuitive System restricts the system's performance on the GSM8K dataset. Conversely, when LLaMA-7B serves as the Intuitive System and GPT2-XL as the Reflective System, the performance improvement on the GSM8K dataset, in comparison to using LLaMA-7B as the Reflective System, is not substantial. This indicates that in this particular case, the Reflective System limits the overall system's performance.

## 4.5 Compared with Other Finetune SOTAs

In pursuit of fairness, we conducted a comparison of state-of-the-art methods that had been fine-tuned on identical datasets, in contrast to our prior evaluation of methods relying on GPT-3.5, which is parameter-free.

**DecomP** (Khot et al., 2023) solves complex tasks by decomposing them (via prompting) into simpler sub-tasks. **Specialize** (Fu et al., 2023) proposes small model specialization to enhance performance by focusing model capacity on a specific target task. **Self-Ask** (Press et al., 2022) explicitly asks the model itself (and then answers) follow-up questions before answering the initial question.

It can be observed in Table 4, when compared with the finetune-based methods, Cogtree still achieves state-of-the-art results. This is attributed to not only decomposing the problem but also incorporating result validation in subsequent steps.

## 4.6 Ablation Study

We conducted further experiments to demonstrate the effectiveness of decomposing step by step. "w/ decomposition" indicates that we used the method from our paper to decompose and sequentially answer the original problem. On the other hand, "w/o decomposition" means that we did not use the intermediate problem decomposition and directly answered the original problem, relying on System 1 to generate the answer.

As we can see in the Table 5, the accuracy of directly answering the original problem is low, especially when the original problem is complex. This is also in line with human cognition. When solving math problems, we also solve intermediate problems first and then obtain the final answer, which improves our accuracy.

## 4.7 Hyper-parameter Analysis

We vary one hyper-parameter with others fixed. From Figure 6, as $\lambda$ increases, the performance first increases and then drops, and it can achieve the best result when $\lambda = 0.5$.

## 5 Related Work

**Multi-step Reasoning.** Reasoning has long been a key focus in natural language processing research. Initially, most studies concentrated on basic tasks like single-sentence language inference (Zamansky et al., 2006; MacCartney and Manning, 2009; Angeli et al., 2016; Hu et al., 2020; Chen et al., 2021)

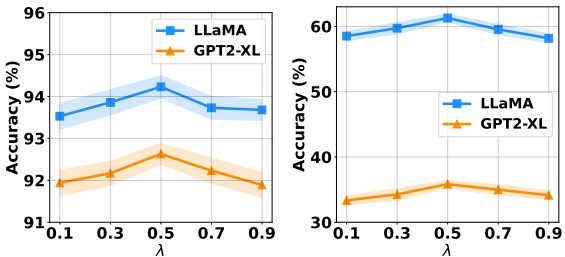

Figure 6: The impact of the hyper-parameter $\lambda$ on EB (left) and GSM8K (right).

and commonsense inference (Rajani et al., 2019; Latcinnik and Berant, 2020; Shwartz et al., 2020) in a single step. Lately, there has been a surge in research interest regarding multi-step reasoning and mathematical problem solving. The search space for a correct reasoning path is extremely vast and complex. Previous approaches (Bostrom et al., 2022; Creswell et al., 2022; Zhao et al., 2023) predominantly emphasized a bottom-up reasoning approach, with a strong focus on system design. In contrast, our approach to constructing the cognitive tree adopts a top-down methodology, reducing the burden during the era of generative models. This top-down approach offers greater flexibility for application across models of varying scales.

**LLM as Evaluation.** The utilization of Language Models (LLMs) to evaluate the validity of their own predictions is gaining significance as a procedure in problem-solving. The introduction of the *self-reflection* mechanism by Shinn et al., 2023; Madaan et al., 2023; Paul et al., 2023 involves LMs providing feedback to their generated candidates. Tree of Thought (Yao et al., 2023) and our approach share a common utilization of a tree-based structure for problem-solving. However, ToT primarily concentrates on tree construction and limited self-validation of intermediate results. According to the dual process theory, which suggests that validation requires deeper levels of thinking, our approach incorporates a contrastive learning method to enhance the model's ability to distinguish accurate results and facilitate comprehensive global validation of the generated outcomes.

## 6 Conclusion

In this paper, we proposed a new framework named CogTree to address complex logical reasoning and mathematical problems. The process of reasoning involves constructing a Cognitive Tree, where the nodes represent the decomposition of complex problems into sub-problems, and the edges represent judgments regarding the correctness of the decomposition. Based on the implementation of our approach over GPT2 and LLaMA, we have achieved results comparable to GPT-3.5 on EB and GSM8K datasets, indicating the effectiveness of our framework.

## Limitations

Due to the limitation of computational resources, we did not test our method on larger scale models. As the model size increases, using our approach may lead to further improvement in the accuracy of answers to these questions. Another direction worth exploring is diversifying the validation methods for the Reflective System, such as using multi-verification to compare the generated results and select the optimal answer.

## Acknowledgements

This work was supported in part by National Natural Science Foundation of China under Grant (No. 62072182, No. 92270119), the Fundamental Research Funds for the Central Universities, and by Alibaba Group through Alibaba Research Intern Program.

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

# A  Appendix

Below we present some successful and failed cases of CogTree for analysis.

**Step=1:**

**Query₁**: the sun will be the star that appears the brightest to the earth

**Decomposition₁₋₁**: [ as the stars become closer, the light of the stars will appear brighter ] + [ the sun is the star that is closest to earth ]

**Decomposition₁₋₂**: [ as the stars become closer, the light of the stars will appear brighter ] + [ a source of something produces that something ]

**Decomposition₁₋₃**: [ a star produces light ] + [ the sun is the star that is closest to earth ]

**Step=2:**

**Query₂**: as the stars become closer, the light of the stars will appear brighter

**Decomposition₂₋₁**: [ stars are a source of light ] + [ as a source of light becomes closer , the light will appear brighter ]

**Decomposition₂₋₂**: [ a source of something produces that something ] + [ stars are a source of light ]

**Decomposition₂₋₃**: [ a star produces light ] + [ a source of something produces that something ]

**Step=3:**

**Query₃**: stars are a source of light

**Decomposition₃₋₁**: [ a star produces light ] + [ a source of something produces that something ]

**Decomposition₃₋₂**: [ a source of something produces that something ] + [ a star produces light ]

**Decomposition₃₋₃**: [ the sun is a star ] + [ a star produces light ]

Figure 7: A successful case in EB of 3 total steps. Green represents answers judged as "sure" by the Reflective System. Gray represents answers judged as "likely" by the Reflective System, and red represents answers judged as "impossible" by the Reflective System.

**Step=1:**

**Query₁**: Natalia sold clips to 48 of her friends in April, and then she sold half as many clips in May. How many clips did Natalia sell altogether in April and May?

**Decomposition₁₋₁**: How many clips did Natalia sell in May? Natalia sold 48/2 = 24 clips in May.

**Decomposition₁₋₂**: How many clips did Natalia sell in April? Natalia sold 48 clips in April.

**Decomposition₁₋₃**: How many clips did Natalia sell in May? Natalia sold 48 clips in May.

**Step=2:**

**Query₂**: : Natalia sold clips to 48 of her friends in April, and then she sold half as many clips in May. How many clips did Natalia sell altogether in April and May? Decomposition₁: How many clips did Natalia sell in May? Natalia sold 48/2 = 24 clips in May.

**Decomposition₂₋₁**: How many clips did Natalia sell altogether in April and May? Natalia sold 48+24 = 72 clips altogether in April and May.

**Decomposition₂₋₂**: How many clips did Natalia sell altogether in April and May? Natalia sold 48+48 = 96 in April and May.

**Decomposition₂₋₃**: How many clips did Natalia sell altogether in April and May? Natalia sold 48 in April and May.

Figure 8: A successful case in GSM8K of 2 total steps. Green represents answers judged as "sure" by the Reflective System. Gray represents answers judged as "likely" by the Reflective System, and red represents answers judged as "impossible" by the Reflective System.

**Step=1:**
Query₁: two tectonic plates pushing toward each other changes the earth's crust material into metamorphic rock

Decomposition₁: [ exposure to intense heat and pressure changes the material of earth's crust into metamorphic rock ] + [ two tectonic plates pushing toward each other cause intense heat and pressure in the earth's crust ]

**Step=2:**
Query₂: exposure to intense heat and pressure changes the material of earth's crust into metamorphic rock

Decomposition₂: [ Exposure to extreme heat and pressure can transform igneous rock into metamorphic rock ] + [ the continental crust mainly is made of granite ]

**Step=3:**
Query₃: exposure to extreme heat and pressure changes the material of earth's crust into metamorphic rock

Decomposition₃: [ exposure to extreme heat and pressure changes the material of earth's crust ] + [ the crust is made of different kinds of crust ]

**Step=4:**
Query₄: exposure to extreme heat and pressure changes the material of earth's crust

Decomposition₄: [ exposure to extreme heat and pressure changes igneous rock into metamorphic rock ] + [ rock is a kind of material ]

**Step=5:**
Query₅: exposure to extreme heat and pressure changes igneous rock into metamorphic rock

Decomposition₅: [ two tectonic plates pushing toward each other cause intense heat and pressure in earth's crust ] + [ rock is a kind of material ]

Figure 9: A failed case in EB of 11 total steps. Our system stops generating after Step 5.

**Step=1:**
Query: John drives for 3 hours at a speed of 60 mph and then turns around because he realizes he forgot something very important at home. He tries to get home in 4 hours but spends the first 2 hours in standstill traffic. He spends the next half-hour driving at a speed of 30mph, before being able to drive the remaining time of the 4 hours going at 80 mph. How far is he from home at the end of those 4 hours?

Decomposition₁: How far was John during the first 3 hours of driving?
John drives at a speed of 60 mph for 3 hours, so the distance covered is calculated as follows: Distance = 60 × 3 = 180 miles

**Step=2:**
Decomposition₂: How far did John driving in the next half hour?
John drives at a speed of 30 mph for half an hour, so the distance covered is calculated as follows: Distance = 30 × 0.5 = 15 miles

**Step=3:**
Decomposition₃: How long is the distance covered during the remaining time of 4 hours?
John spends 2 hours in standstill traffic, so he has 4 hours - 2 hours = 2 hours remaining to drive at a speed of 80 mph. Distance = 80 × 2 = 160 miles

**Step=4:**
Decomposition₄: How long is the total distance covered during the 4 hours?
To find the total distance covered, we add up the distances calculated in the previous steps: Total Distance = 180 + 15 + 160 = 355 miles

Figure 10: A failed case in GSM8K of 6 total steps. Our system stops generating after Step 4.