# OpenReview forum: "From Complex to Simple: Unraveling the Cognitive Tree for Reasoning with Small Language Models"
_EMNLP/2023/Conference — EMNLP 2023 Findings_

### Official Review · Reviewer_UpNQ · 2023-08-03

**Soundness:** 3

**Excitement:**

4: Strong: This paper deepens the understanding of some phenomenon or lowers the barriers to an existing research direction.

**Paper Topic And Main Contributions:**

This paper proposes Cognitive Tree, a methodology to enhance the capatibility of LLMs in solving logical reasoning. It draws inspiration from the two-systems theory in cognitive science: the implicit extraction module (intuitive system), which generates a set of different potential answers to the query; the explicit reasoning module (the reflective system) which compares and scores these answers in order to obtain the right one. The results of the experiments show that this approach improves the system's efficiency by reducing the number of mecessary paramenters to answer the query.

**Questions For The Authors:**

- How the sub-questions are generated? Section 3.1 is not clear about that. Although Figure 3 and Figure 4 show examples, it's difficult to understand how the decomposition is generated due to significant differences between them.
- Regarding the ending step, the caption in Figure 2 says, "the process terminates when the decomposition successfully matches the target answer". I assume that this is just the for training step, isn't it?

**Reasons To Accept:**

- The paper addresses two relevant issues: the low performance of LLMs in reasoning problems and their computational costs.
- It draws inspiration from cognitive science, making it easier to understand. Additionally, it applies contrastive learning, where reasoning steps and negative results play a major role which makes the system more understandable.
- The experiments seem to be well-designed and robust.
- The results enhance the accuracy of the systems and, more importantly, significantly reduce the number of parameters required for the calculation, although it requires more intermediate steps.


**Reasons To Reject:**

- A key point of the method, the decomposition of the queries, is not explained in detail, making very difficult to understand how the sub-questions are generated. In section 3.3 is mentioned "by leveraging in-context examples", but it's not clear to me what this means.

**Reproducibility:**

3: Could reproduce the results with some difficulty. The settings of parameters are underspecified or subjectively determined; the training/evaluation data are not widely available.

**Reviewer Confidence:**

2: Willing to defend my evaluation, but it is fairly likely that I missed some details, didn't understand some central points, or can't be sure about the novelty of the work.

**Typos Grammar Style And Presentation Improvements:**

- Daniel Khaneman reference is wrong.
- Line 333: missing reference

---

> ### Author Rebuttal · Authors · 2023-08-27
>
> ### Q1: How the sub-questions are generated? Section 3.1 is not clear about that. Although Figure 3 and Figure 4 show examples, it's difficult to understand how the decomposition is generated due to significant differences between them.
> Our dataset includes the decomposition of each question. ( \[ Query_1: Decomposition_1, Query_2: Decomposition_2, ... , Query_n: Decomposition_n \] )
>
> For math problems, the decomposition is like:
>
> Query_1: Natalia sold clips to 48 of her friends in April, and then she sold half as many clips in May. How many clips did Natalia sell altogether in April and May?
>
> Decomposition_1: How many clips did Natalia sell in April? Natalia sold 48 clips in April.
>
> Decomposition_2: How many clips did Natalia sell in May? Natalia sold 48/2 = 24 clips in May.
>
> For logical reasoning, the decomposition is like:
>
> Query1: earth rotating on its axis causes cycles of day and night on earth
>
> Decomposition1: \[ earth is a planet that rotates on its tilted axis \] + \[ a planet rotating causes cycles of day and night on that planet \]
>
> Therefore, we can use the decomposition of other queries as context for the current query. For logical and mathematical data, our training is isolated. This means that mathematical and logical data are not trained together. In our training process, we select K examples as context to guide the model to generate the decomposition for the current problem. For example:
>
> Query_1: xxxx
>
> Decomposition_1: xxx
>
> Query_2: xxxx
>
> Decomposition_2: xxx
>
> Query_3: xxxx
>
> Decomposition_3: xxx
>
> Current Query: xxx
>
> Current Decomposition: To be generated
>
> ### Q2: Regarding the ending step, the caption in Figure 2 says, "the process terminates when the decomposition successfully matches the target answer". I assume that this is just the for training step, isn't it?
> In fact, it is during the inference stage that we need to determine whether the process terminates (lines 342-350). We add an end token "End" after each question during the training stage, and when the generation system reaches the "End" token or when the maximum iteration limit of 20 is reached, our framework stops the reasoning process.
>
> We sincerely thank for your time and efforts.

---

### Official Review · Reviewer_BP1d · 2023-08-04

**Soundness:** 4

**Excitement:**

4: Strong: This paper deepens the understanding of some phenomenon or lowers the barriers to an existing research direction.

**Paper Topic And Main Contributions:**

This paper proposes a novel framework (CogTree) aimed at building a tree-like structure going all throughout from a query (root node) to the leaf nodes, containing questions that need to be answered.
The question whether LMs are able to gain any sort of semantic knowledge and logical reasoning ability is a relevant one, so that paper is fully appropriate for the Audience of this Conference.

The paper nicely starts with a reference to Cognitive Science literature, namely to the dual process theory: one chief trait of the proposed work lies in the distinction between an implicit extraction module (which is clearly to be mapped onto 'system one' processes), and a reflective module (which should be framed as a 'system two' set of processes).


A state-space search approach is adopted to break down the goal into smaller components, that allow reaching the goal (which is represented as a query). At the heart of the reflective system we have a prompt-based approach, where the model is employed to output a judgement on the appropriateness between query and its decompositions on three classes: sure, impossible or likely.

The evaluation was conducted on the Entailment Bank (EB)and the GSM8K dataset, and employing GPT2-XL and LLaMA-7B models. Experimental results show that a lighter LM (GPT2-XL, 1.5B) outperforms a richer LM (GPT-3.5, 175B). The latter model obtains improved accuracies with Chain-of-though and Tree-of-thought prompting.


The work is well structured and well written; it provides enough detail to grasp all main elements (the code will be released upon acceptance).

I have one main concern. At their core, LM are basically probability distributions over language vocabularies: as such, LMs are unfit to deal with logical reasoning. They can be trained with sophisticated objectives, but still this will result in getting (only) more accurate distributions while reasoning is a matter of propositional knowledge. I am sure that the Authors will follow this argument: seeing that Weng earns $12 per hour for babysitting, and that yesterday she just did 50 minutes of babysitting, thereby earning $10 is almost useless to make a decision on the case of Frida, who earns $500 per (5 working days a) week but worked only three days this week. The underlying computational device is similar, but by observing sentences and tokens herein no suitable generalization can be drawn from the case of Weng to that of Frida. Can the Authors explain how a LM can be able in principle to generalize from the case of Weng to another one, such as the case of Frida?



POSSIBLE IMPROVEMENTS
- lines 209-19 (on the Decomposition step) are unclear, should be rephrased to clarify (illustrating scores associated to a working examples would be beneficial)
- Figure 5: some description for the x axis must be provided, e.g. 'inference chain length'
- the content of the Limitations section has not to do with actual limitations: en effort to elaborate on limitations is required here...


**Questions For The Authors:**

[Q1] At their core, LM are basically probability distributions over language vocabularies: as such, LMs are unfit to deal with logical reasoning. They can be trained with sophisticated objectives, but still this will result in getting (only) more accurate distributions while reasoning is a matter of propositional knowledge. I am sure that the Authors will follow this argument: seeing that Weng earns $12 per hour for babysitting, and that yesterday she just did 50 minutes of babysitting, thereby earning $10 is almost useless to make a decision on the case of Frida, who earns $500 per (5 working days a) week but worked only three days this week. The underlying computational device is similar, but by observing sentences and tokens herein no suitable generalization can be drawn from the case of Weng to that of Frida. Can the Authors explain how a LM can be able in principle to generalize from the case of Weng to another one, such as the case of Frida?


**Reasons To Accept:**

The question whether LMs are able to gain any sort of semantic knowledge and logical reasoning ability is a relevant one, so that paper is fully appropriate for the Audience of this Conference.

**Reasons To Reject:**

- It seems that LMs are not fully adequate to deal with higher-order tasks, such as complex logical reasoning. Their usage should thus be supported by some sort of either theoretical or practical motivation.

**Reproducibility:**

4: Could mostly reproduce the results, but there may be some variation because of sample variance or minor variations in their interpretation of the protocol or method.

**Reviewer Confidence:**

4: Quite sure. I tried to check the important points carefully. It's unlikely, though conceivable, that I missed something that should affect my ratings.

**Typos Grammar Style And Presentation Improvements:**


Minor issues & Typos
- line 082: citation 'Daniel, 2017' should be corrected into 'Kahneman, 2017';
- line 333: missing reference LLaMA-7B (?)
- line 357: reasoning abilitiy -> ability
- Figure 5 is too tiny, nearly unreadable, should anybody ever try to print the paper...

---

> ### Author Rebuttal · Authors · 2023-08-27
>
> ### Q1: Can the Authors explain how a LM can be able in principle to generalize from the case of Weng to another one, such as the case of Frida?
> The generation at the token level does indeed impose limitations on the language model's performance in tasks involving logical reasoning and propositional tasks. However, what is surprising is that despite the token-level training of large models, such as GPT-4, they possess the ability to perform logical reasoning and provide mathematical answers. One of OpenAI's findings is that code pre-training aids in enhancing the reasoning capabilities of these large models. The rationale behind this lies in the fact that code inherently contains a wealth of logical judgments, with interconnections between code blocks. Similarly, in natural language, there exists a corpus of expressions like "because XXX, therefore XXX". Pre-training on such materials equips language models with reasoning abilities for downstream tasks (as downstream tasks essentially reuse the abilities obtained during pre-training).
>
> Returning to your question, "how can the language model generalize from the case of Weng to that of Frida?" Our understanding is that the model, after being exposed to extensive pre-training data, acquires the capability to generalize from one category of logical reasoning task to another similar task. Fine-tuning in the downstream phase activates this specific task-related ability. One potential improvement in the future could involve generating potential choices for the next paragraph, rather than predicting the next token, as this might enhance the language model's problem-solving and logical reasoning capabilities.
>
> We sincerely thank for your time and efforts.

---

### Official Review · Reviewer_gKx3 · 2023-08-05

**Typos Grammar Style And Presentation Improvements:** 1. in line 333, reference error.
**Soundness:** 2

**Excitement:**

2: Mediocre: This paper makes marginal contributions (vs non-contemporaneous work), so I would rather not see it in the conference.

**Missing References:**

@misc{shridhar2023distilling,
      title={Distilling Reasoning Capabilities into Smaller Language Models},
      author={Kumar Shridhar and Alessandro Stolfo and Mrinmaya Sachan},
      year={2023},
      eprint={2212.00193},
      archivePrefix={arXiv},
      primaryClass={cs.LG}
}

@misc{zhou2022learning,
      title={Learning to Decompose: Hypothetical Question Decomposition Based on Comparable Texts},
      author={Ben Zhou and Kyle Richardson and Xiaodong Yu and Dan Roth},
      year={2022},
      eprint={2210.16865},
      archivePrefix={arXiv},
      primaryClass={cs.CL}
}

**Paper Topic And Main Contributions:**

This paper introduces CogTree, a system method constitute of a generator and a discriminator, which are applied to decomposed steps in reasoning tasks.

Main contributions include a comparable performance with GPT-3.5 using smaller LMs.

**Questions For The Authors:**

1. Most training datasets do not contain intermediate subquestions and answers. Is CogTree invalid in such situations?
2. CogTree is finetuned on the same distribution as the test set (EB and GSM8K), is it unfair to compare with LLMs which are few-shots? Expected to see the experiments compared with other finetune SOTAs.
3. Why decomposing step by step? Does the experiment prove the effectiveness of decomposing?

**Reasons To Accept:**

1. A thorough experiment about the CogTree method. Prove the effectiveness of finetuning and adding a reflective system.

**Reasons To Reject:**

1. Although the story of the intuitive and reflective system sounds fancy, essentially, it's a traditional system with a generator along with a scoring mechanism (discriminator). The difference lies in the decomposition part, which was discussed widely in previous work like Least-to-Most prompting, Self-Ask,  Decomposition Distillation, DecompT5, etc.
2. Expected to see the performance gain compared with other SOTA finetune-based methods.
3. Expected to see a general method to adapt to datasets that do not have intermediate subquestions and answers, which is common in most situations.

**Reproducibility:**

4: Could mostly reproduce the results, but there may be some variation because of sample variance or minor variations in their interpretation of the protocol or method.

**Reviewer Confidence:**

4: Quite sure. I tried to check the important points carefully. It's unlikely, though conceivable, that I missed something that should affect my ratings.

---

> ### Author Rebuttal · Authors · 2023-08-27
>
> ### Q1: Most training datasets do not contain intermediate subquestions and answers. Is CogTree invalid in such situations?
> A significant distinction of our approach from other decomposition methods lies in the fact that, while previous methods such as Least-to-Most prompting and DecompT5 focus on how to break down problems and utilize intermediate results, our approach not only generates multiple decomposition hypotheses for the current problem but also validates the intermediate results of the decomposition. This enhances the accuracy of multi-step reasoning leading to the final result. In our experiments, we adhere to the established settings from previous papers. In fact, our approach follows the standard settings of LLM reasoning, including Chain-of-Thought (CoT), Least-to-Most Prompting, and Tree-of-Thought (ToT). These papers on LLM reasoning still rely on intermediate-step results for inference to arrive at the final answer. While a method that can directly solve complex reasoning problems without intermediate steps is attractive, it remains challenging for the current LLMs. Consider the way humans solve problems: when tackling a complex problem, we break it down into smaller components to solve the original problem. Without intermediate steps, our error rate would also be quite high. The essence of this problem is whether our model can generalize to a dataset without intermediate problem decomposition through training on datasets that involve intermediate steps, and correctly decompose and answer them. This is a direction we can improve in the future. However, currently, we can provide a semi-automatic approach: for a new class of data or problems, we can manually generate subproblems and their answers, use this content as context input to LLM or similar models like ChatGPT, and let it generate the subsequent steps as training data for the smaller model.
>
> ### Q2: CogTree is finetuned on the same distribution as the test set (EB and GSM8K), is it unfair to compare with LLMs which are few-shots? Expected to see the experiments compared with other finetune SOTAs.
> We further compared the performance of our method with other state-of-the-art methods fine-tuned on the datasets, including:
>
> - **DecomP**: Decomposed Prompting: A Modular Approach for Solving Complex Tasks
> - **Specialize**: Specializing Smaller Language Models towards Multi-Step Reasoning
> - **Self-Ask**: Measuring and Narrowing the Compositionality Gap in Language Models
>
> | Model        | EB    | GSM8K |
> |--------------|-------|-------|
> | GPT2-XL(1.5B) |       |       |
> | +Specialize  | 81.54 | 21.43 |
> | +DecomP      | 83.27 | 24.35 |
> | +Self-Ask    | 82.69 | 25.46 |
> | +CogTree     | 93.16 | 34.68 |
> | Llama-7B     |       |       |
> | +Specialize  | 84.32 | 34.21 |
> | +DecomP      | 82.43 | 39.75 |
> | +Self-Ask    | 83.98 | 38.35 |
> | +CogTree     | 94.25 | 61.28 |
>
> *Table*: Performance metrics for different methods with EB and GSM8K.
>
> It can be observed that, when compared with the finetune-based methods, Cogtree still achieves state-of-the-art results. This is attributed to not only decomposing the problem but also incorporating result validation in subsequent steps.
>
> ### Q3: Why decomposing step by step? Does the experiment prove the effectiveness of decomposing?
> Step-by-step decomposition refers to breaking down a complex problem into simpler problems, and the effectiveness of this approach has been demonstrated in Least-to-Most. We conducted additional experiments. "w/ decomposition" indicates that we used the method from our paper to decompose and sequentially answer the original problem. On the other hand, "w/o decomposition" means that we did not use the intermediate problem decomposition and directly answered the original problem, relying on System 1 to generate the answer.
>
> | Model   | w/ decomposition EB | w/o decomposition EB | w/ decomposition GSM8K | w/o decomposition GSM8K |
> |---------|----------------------|----------------------|-----------------------|-----------------------|
> | GPT2-XL    | 92.63                | 35.84                | 62.38                 | 18.94                 |
> | LLama-7B  | 93.25                | 61.28                | 69.79                 | 27.16                 |
>
>
> *Table*: Comparison of models with and without decomposition.
>
> As we can see, the accuracy of directly answering the original problem is low, especially when the original problem is complex. This is also in line with human cognition. When solving math problems, we also solve intermediate problems first and then obtain the final answer, which improves our accuracy.
>
> ### Missing References
> Sorry for missing these two citations. We'll add them in the revised version.
>
> We sincerely thank for your time and efforts.

---

### Meta-Review · Area_Chair_aj5Z · 2023-09-11

**Recommendation:** 3

**Metareview:**

This paper presents a methodology for approaching complex commonsense question answering via decomposition, and component-wise verification. The problem is of high significance to the NLP community, and the story builds on a catchy link to cognitive theory by Kahnemann et al. (system 1/system 2).

The paper generally presents a sound piece of work, but there are concerns over the novelty of the contribution, in particular, the abstract contains a very sweeping claim ("we are the first to unravel the cognitive reasoning abilities of LMs"), while several related works on question decomposition are not at all, or insufficiently, discussed. Similarly, the explanation of the core of the methodology, the generation of subquestions, is not entirely clear, and would benefit from step-by-step discussion of a running example.

Overall, the significance of the topic, and presented ideas warrant publication, yet the paper would significantly benefit from a careful revision, in particular w.r.t. its positioning.

---

### Decision · Program_Chairs · 2023-10-07

**Decision:**

Accept-Findings

**Comment:**

This paper presents a methodology for approaching complex commonsense question answering via decomposition, and component-wise verification. The problem is of high significance to the NLP community, and the story builds on a catchy link to cognitive theory by Kahnemann et al. (system 1/system 2).

The paper generally presents a sound piece of work, but there are concerns over the novelty of the contribution, in particular, the abstract contains a very sweeping claim ("we are the first to unravel the cognitive reasoning abilities of LMs"), while several related works on question decomposition are not at all, or insufficiently, discussed. Similarly, the explanation of the core of the methodology, the generation of subquestions, is not entirely clear, and would benefit from step-by-step discussion of a running example.

Overall, the significance of the topic, and presented ideas warrant publication, yet the paper would significantly benefit from a careful revision, in particular w.r.t. its positioning.